# Hybrid Renewable Production Scheduling for a PV–Wind-EV-Battery Architecture Using Sequential Quadratic Programming and Long Short-Term Memory–K-Nearest Neighbors Learning for Smart Buildings

**Asmae Chakir and Mohamed Tabaa ***

Pluridisciplinary Laboratory of Research & Innovation (LPRI), Moroccan School of Engineering, Casablanca 20220, Morocco; a.chakir@emsi.ma

\* Correspondence: a.tabaa@emsi.ma

**Abstract:** Electricity demand in residential areas is generally met by the local low-voltage grid or, alternatively, the national grid, which produces electricity using thermal power stations based on conventional sources. These generators are holding back the revolution and the transition to a green planet, being unable to cope with climatic constraints. In the residential context, to ensure a smooth transition to an ecological green city, the idea of using alternative sources will offer the solution. These alternatives must be renewable and naturally available on the planet. This requires a generation that is very responsive to the constraints of the 21st century. However, these sources are intermittent and require a hybrid solution known as Hybrid Renewable Energy Systems (HRESs). To this end, we have designed a hybrid system based on PV-, wind-turbine- and grid-supported battery storage and an electric vehicle connected to a residential building. We proposed an energy management system based on nonlinear programming. This optimization was solved using sequential quadrature programming. The data were then processed using a long short-term memory (LSTM) model to predict, with the contribution and cooperation of each source, how to meet the energy needs of each home. The prediction was ensured with an accuracy of around 95%. These prediction results have been injected into K-nearest neighbors (KNN), random forest (RF) and gradient boost (GRU) repressors to predict the storage collaboration rates handled by the local battery and the electric vehicle. Results have shown an R2_score of 0.6953, 0.8381, and 0.739, respectively. This combination permitted an efficient prediction of the potential consumption from the grid with a value of an $R^2$-score of around 0.9834 using LSTM. This methodology is effective in allowing us to know in advance the amount of energy of each source, storage, and excess grid injection and to propose the switching control of the hybrid architecture.

**Keywords:** renewable energy; energy management; nonlinear programming; sequential quadrature programming; hybrid architecture; machine learning; LSTM prediction; random forest regression; K-nearest neighbors; gradient boost regression

## 1. Introduction

The population and the living standard development of the inhabitants raise the energy consumption in a city. This increase imposes a certain proportionality of the energy production, obviously to satisfy the electricity request. The energy production must follow an intelligent and modern process to meet the requirements of the century. Indeed, from now on, the production must be efficient and environmentally friendly, especially with the climate change effect and the expensive fuel prices [1,2]. Due to these circumstances, renewable sources have allowed an advancement in the energy sector that can be considered very beneficial [3]. The use of these renewable sources has been carried out on a large scale [4]. This solution obviously ensures a large amount of production but

follows an on-and-off production system. This rule is equivalent to having production if the renewable potential in question is present and a non-existent production in the opposite case. This is explained by the intermittent nature of the renewable sources. To overcome this problem, hybrid systems based on complementary renewable sources have been adopted as a solution [5–7]. Indeed, once one of the sources is no longer producible, the other source takes over, and vice versa. Several hybridizations are possible under the condition of complementarity. Various research studies have been conducted in this direction and many hybrid systems have been studied. The PV–wind is the system that is most compatible for areas with diverse climates such as Morocco [8,9]. However, other sources could be used if the renewable potential allows it. Indeed, hydraulic microturbines could be a solution to empower the hybrid system in an efficient way, especially for direct addition to the utility grid and even of self-consumption, in particular applications [10]. To further smooth the production profile of renewable systems, a storage system is added, whose type depends on the application constraints. By means of this hybridization, we can tackle the increasing demand for electricity in the residential sector following a small-scale version. This system's size will be limited by the building's overall occupancy and the loads to be met. Alongside the residential sector, we find the transportation sector, which also requires a high power level to satisfy the transition towards electric transportation. This implies considering local loads within the home and is a good solution to further minimize reliance on the national grid. So, the system in question will feature a very powerful storage system. Indeed, local storage via the lead–acid battery, mobile storage via the electric car, and finally infinite-capacity storage via the national grid are possibilities [11].

Even if hybridization solves problems such as intermittency, it requires several research studies into, namely, the maximum power point tracking (MPPT) and the sizing and the management of the produced renewable energy, especially since it is hybrid. The maximum power extraction is an important issue for renewable sources because they represent systems which, during their normal operation, are not located at the point of peak power extraction. For this reason, several methods have been developed to solve this problem, either traditionally or based on artificial intelligence, both for solar and wind systems [12,13]. However, before considering the extraction of maximum power, it is necessary to size the renewable system, the so-called hybrid one. Hybridizing electrical production systems, whether renewable or conventional, generates issues such as sizing the system and the approach to be followed for optimal resource use [14,15]. Obviously, the system must respond to an application of the solution to these potential constraints, noting, among other factors, the spatial area. Since the aim is to create a hybrid system, it is important to optimize the overall system design. Indeed, a hybrid system based on renewable energy contains at least two sources managed by two distinct renewable potentials. Consequently, using a hybrid system means optimally matching the combination size to meet a given energy supply requirement.

The last and the most important aspect required to take advantage of the first two optimization phases is the hybrid system's energy management [16]. Like sizing, the hybrid power system's energy management involves how and when to use the renewable energy available at a given time in the generation chain. To this end, authors have used several methods to manage the energy produced by a hybrid power system, whether using rule-based methods, meta-heuristic methods, or straightforward linear or non-linear mathematical programming [17,18]. Indeed, in [19] the authors have proposed a rule-based energy algorithm to manage the supply of a household already connected to the grid equipped with a PV-battery renewable system. Others in [20] have proposed an energy management system based on a linear programming model to schedule the energy produced by a PV–wind–battery hybrid system. In [21], they have developed a new method based on parallel hybrid genetic algorithm–particle swarm optimization algorithms to size and manage the energy used to support the integration of renewable energy in Laayoune, a Moroccan city. The concept of artificial intelligence has made energy management a more responsive and efficient research field. Indeed, in [22] the authors have used a time-delay

neural network combined with stochastic model predictive control for a proper energy management system for renewable source communities. Others in [23] have combined reinforcement learning with a neural network for smart-home energy management to find the best time for energy use. The use of artificial intelligence in the renewable energy field helped to attend rapidly and widely to the sustainable development goals. Indeed, until now, artificial intelligence has helped the renewable energy to achieve 42 out of 169 sustainable development targets [24], but the development and investigations will increase the achievement, and more targets could be tackled, with the renewable energy combined with artificial intelligence. By using decision trees, the authors in [25] have developed a process to improve the monitoring systems in smart buildings that help with energy distribution efficiency. The authors in [26] have proposed a comprehensive method to optimize the renewable energy production based on a hybrid LSTM-RL model in smart-grid application. The authors in [27] found that GRU is the most suitable to predict the output of wind turbine production compared with a statistical method. The forecasting does not identify the traits of just the renewable production in the residential sector specifically, but also the thermal loads in a smart building. The authors in [28] have treated this area by using the fuzzy radial basis function neural network to forecast the building loads. In [29], the authors have captured features and studied the forecasting of a system with multiple sources based on a convolutional neural network and attention-based long short-term memory.

According to the literature, even if we have a hybrid system based on several renewable energies, the extraction of maximum power is carried out according to each separate source. Conversely, a hybrid system is designed to respond in a hybrid way to an energy demand, whereas making a separate decision for each source risks reducing a system's efficiency. This proves the ultimate utility of the energy management section; hence our contribution. In this paper, we therefore propose to combine a typically sized renewable hybrid system with an energy management system that will serve directly to meet the demand of a residential building equipped with an electric vehicle charging terminal. The system has been designed to suit the most extreme weather conditions, and the battery storage system ensures three days' autonomy for all the household electrical loads, without having to rely on renewable sources or the national grid. To do so, for a hybrid system based on PV–wind–EV-battery, a nonlinear programming combined with LSTM-KNN is proposed. Non-linear programming is used to plan system interactions using quadratic sequential resolution, which will then help with the energy management system in order to control effectively the production and the consumption of the system. The results are then processed with long short-term memory (LSTM) combined with K-nearest neighbors (KNN) to predict the following: management sequences that will be useful for management of the energy produced or consumed by the hybrid system and the household, control of the DC/DC, AC/DC and DC/AC power electronic converters used to meet the common bus requirements, and the switching sequences of different interrupters and switches of the hybrid system architecture.

This paper is organized as follows. The first section is dedicated to the hybrid system modeling. The second section describes the methodology used to model the proposed optimal management system. The results and discussion are then presented. The conclusion is drawn at the end.

## 2. Hybrid System Modeling

We consider in this study a hybrid architecture of a configuration as clarified in Figure 1. The architecture is based on a hybrid system with two sources, namely the wind turbine and the photovoltaic system. As the renewable hybrid system comes to reduce the load on a single-source renewable generation set, the battery storage system comes to smooth the generation during the periods when the renewable potential is absent. Therefore, the system in question is a PV–wind–EV-battery one. This is carried out by connecting the system to the building through a DC bus. This configuration is built by the structure presented

in Figure 1. There is also a facilitation of the energy use through the development of an energy management system. This later ensures that the hybrid sources contribute equally to supplying the different electrical loads installed inside the house and cooperate with the neighborhood electrical micro-grid via the surplus production. The management system takes into consideration the variations and specifications of each renewable source, as well as the state of charge of the battery storage system and the electric vehicle availability. The switches shown in the figure ensure an energy control facility. Indeed, S1 and S5 control the collaboration of the solar system with the DC bus and the injection of its production to the grid. S2 and S6 manage the production on the DC bus and the injection to the grid of the wind system energy. S3 represents the bi-directional collaboration of the battery. S4 is for the grid collaboration in the hybrid configuration. Finally, the switch S7 provides for the bi-directional collaboration of the electric vehicle. Moreover, a further criterion that should be considered by our control system is the electrical load category to be served, as classified in Figure 2. In our case, four types of loads are involved. Firstly, we find the uncontrollable loads, which are directly related to the user's comfort and cannot be controlled without the risk of disturbing or reducing the comfort expected by the users. Secondly, we find the reducible loads, which are tolerable in terms of power. These loads can be scaled down to meet an energy shortage. Thirdly, we have the loads that can be switched off to meet an energy shortage. This type of load is divided into two main types: the loads that are interruptible, which tolerate load shedding at any time, and others that are non-interruptible, since they work according to load cycles. Once they are interrupted in the middle of their work, there is a risk that they will start again from their first cycle, which means consuming more energy.

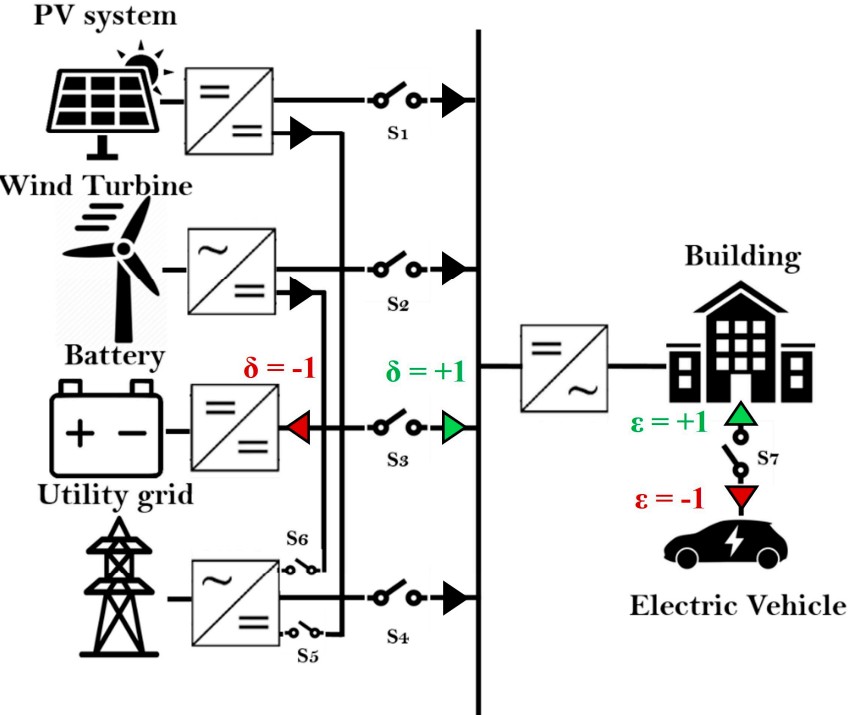

**Figure 1.** Hybrid renewable-energy- system configuration.

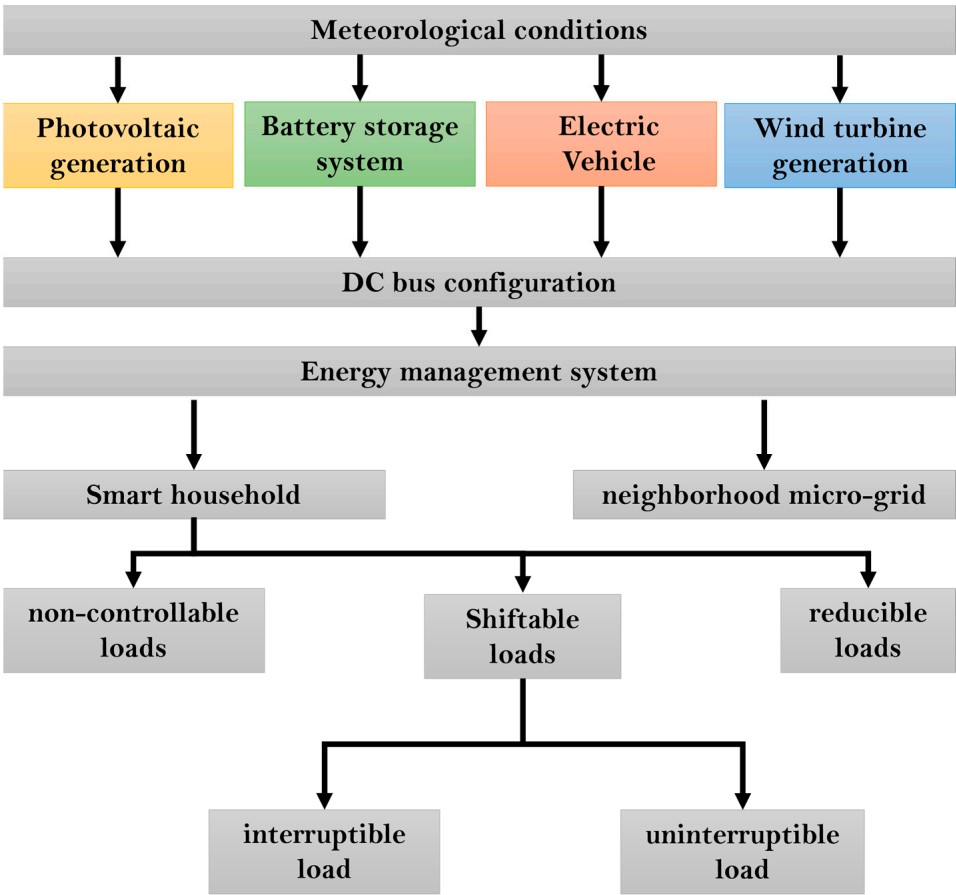

**Figure 2.** Hybrid architecture configuration.

### 2.1. Photovoltaic Generation

Photovoltaic generation is based on solar panels made of solar cells. The cells are made of a semiconductor material. Based on the solar rays, an electric field is created which allows the conversion of light into electrical energy in a continuous mode [30]. This energy form is ready to be injected into the DC bus. The form of the power produced by the panels is direct current power, $P_{PVMPP}$. Therefore, it is calculated by multiplying the output current $I_{PVMPP}$, and voltage $V_{PVMPP}$ in their continuous form. The system requires power electronic converters for the regulation of the DC bus voltage, as well as for the extraction of maximum power. Therefore, our modeling accounts for the values of the variables in line with their maximum values; see Equation (1).

$$P_{PVMPP} = I_{PVMPP} \times V_{PVMPP} \qquad (1)$$

The output current of the panel is divided into three flows: the photocurrent, the current flowing through the diode, and the leaking current passing through the leakage resistor [31], according to a mono-diode equivalent scheme. All these types of current are a matter of meteorological conditions, namely, the ambient temperature T and the ambient irradiations noted as G, except for the current that passes through the leakage resistor, which is a function of the output voltage of the panel. As a result, the total power produced by the panel is the voltage function, which is calculated according to a maximum power point, noted as $P_{PVMPP}$ (G, T, $V_{PVMPP}$). This voltage was estimated using the Lambert W function [19], as mentioned below:

$$B(T) = \frac{(I_{scn} + k_i(T - T_n))N_p}{\exp\left(\frac{(V_{ocn} + k_v(T - T_n))N_{ss}}{\frac{nN_s k_B TN_{ss}}{q}}\right) - 1} \tag{2}$$

$$A(G,T) = \frac{G}{G_n}[I_{scn} + k_i(T - T_n)]N_p \tag{3}$$

$$P_{PVMPP}(T,G) = \frac{nN_s k_B TN_{ss}}{q}\left(A(T,G) - B(T,G)\frac{qV_{PVMPP}(t)}{nN_s k_B TN_{ss}}\right)\left(\frac{qV_{PVMPP}(t)}{nN_s k_B TN_{ss}} - 1\right) \times \eta_{DC/DC} \tag{4}$$

where, $T_n$ and $G_n$ are the temperature and irradiation of standard conditions, namely, 25 °C and 1000 W/m². $I_{scn}$ is the short current produced during $(T_n, G_n)$ and $k_i$ is its coefficient. $V_{ocn}$ is the open-circuit-voltage produced during $(T_n, G_n)$ and $k_v$ is its coefficient. $K_B$, q and n are the Boltzmann constant, the electron charge and the diode ideality factor, respectively. $N_s$, $N_p$, $N_{ss}$ are the number of PV cells connected in series and the number of panels connected in parallel and in series, respectively. Finally, the efficiency of the DC/DC converter is counted by the coefficient $\eta_{DC/DC}$.

## 2.2. Wind Turbine Generation

The energy captured by the blades of the wind turbine is first converted into mechanical energy before it is converted into electrical energy in an alternative form [32,33]. The aerodynamic shape influences the way the wind turbine captures energy, as well as the efficiency. However, wind turbines, up to this point in time, are unable to draw more energy than the Betz limit of 16/27, estimated at 59% [34,35]. This is due to the aerodynamic shape, no matter what type of generator is connected to the turbine. In our case, we study a horizontal-axis wind turbine with three blades. Its modeling is considered by the following equations [36], where $P_{WTMPP}$ is the wind turbine power produced during optimal control. S, $\rho$ and $V_{Wind}$ are the blade-swept area equal to $\pi R^2$, air density and linear wind speed, respectively. $C_{p-opt}$ is the maximum value reached by the aerodynamic style of the chosen wind turbine. The efficiency of the power electronic converter is estimated by the coefficient $\eta_{AC/DC}$.

$$P_{WTMPP} = \frac{1}{2} \times \rho \times S \times C_{p-opt} \times V_{wind}^3 \times \eta_{AC/DC} \tag{5}$$

$$C_{p-opt} = 0.517\left(\frac{116}{\lambda_{opt} + 0.08\beta_{opt}} - \frac{4.06}{1 + \beta_{opt}^3} - 0.8\beta_{opt} - 5\right)e^{\left(-\frac{21}{\lambda_i}\right)} + 0.0068\lambda_{opt} \tag{6}$$

$$\frac{1}{\lambda_i} = \frac{1}{\lambda_{opt} + 0.08\beta_{opt}} - \frac{0.035}{\beta_{opt}^3 + 1} \tag{7}$$

$$\lambda_{opt} = \frac{R\Omega}{V_{wind}} \tag{8}$$

The optimal power coefficient value is calculated by identifying the result of the pitch angle $\beta$ and the tip-speed ratio $\lambda$ optimal values, noted as $\beta_{opt}$ and $\lambda_{opt}$, respectively, where R is the blade radius and $\Omega$ is the angular velocity.

## 2.3. Electric Vehicle

The storage system in its mobile form is ensured by the electric vehicle. The charging of this battery could be carried out outdoor during the vehicle trip from the grid or renewable charging stations. Otherwise, indoor charging is ensured by the local hybrid renewable system. The vehicle could be used in vehicle-to-home (V2H) or home-to-vehicle (H2V) modes.

$$E_{EV}(t) = E_{EV}(t-1) + \Delta E_{EV,V/H} - E_{EV,Trip} \tag{9}$$

$$\Delta E_{EV,V/H} = \begin{cases} P_{EV} \times \eta_{ch} \times \eta_{AC/DC} & \text{if } \varepsilon > 0 \\ \frac{P_{EV}}{\eta_{dis} \times \eta_{AC/DC}} & \text{if } \varepsilon < 0 \\ 0 & \text{if } \varepsilon = 0 \end{cases} \tag{10}$$

$$E_{EV,Trip} = \frac{1}{2} m \left( \frac{d_{trip}}{\Delta t} \right)^2 \tag{11}$$

$$SOCEV_{min} \leq SoCEV(t) \leq SOCEV_{max} \tag{12}$$

The energy variation of the electric vehicle is then estimated using Equation (9), where $E_{EV}(t)$ and $E_{EV}(t-1)$ represent the energy of the vehicle during the instants t and $t-1$, respectively. The interaction between the home and the vehicle is represented by $\Delta E_{EV,V/H}$. This interaction depends on the power of the electric vehicle. The charging mode is activated when the power available for the electric vehicle in the bus is negative; otherwise, the discharging is activated when $P_{EV}$ is positive. In both charging and discharging modes, the modeling takes into consideration the discharging and charging battery efficiency, represented by $\eta_{dis}$ and $\eta_{ch}$, respectively. Otherwise, the energy consumed during the trip is estimated by $E_{EV,Trip}$, and developed in Equation (11). In fact, the amount of energy consumed during a trip depends on the distance of the trip $d_{trip}$ and the time spent, represented by $\Delta t$ in Equation (11). m is the mass of the car with its driver. The challenges that vehicle systems can face are first and foremost the management of energy flows, which has been remedied with the proposal of our advanced management system, which considers the vehicle as a mobile storage entity. This means that the vehicle will not be charged at home all the time, using the grid during off-peak hours for example, but will also store surplus energy from renewable energies, as well as charging stations away from home when traveling. We also note the compatibility issue with charging terminals, which has been addressed by an AC integration at home level using a Type 2 AC. The final point is the communication and data-exchange difficulties, which can be remedied by standardizing protocols such as ISO15118 [37] for plug and charge.

### 2.4. Battery Storage System

The storage system, regardless of its technology, is very important, especially for renewable energies, which are characterized by the criterion of intermittence [38,39]. In our case, we selected a storage system based on batteries. This type of storage has a state identification parameter that is crucial, namely, the state of charge, noted as SOC. This parameter is none other than the ratio of the energy available to be shared by the battery and the total energy that it allows to be stored. This is shown by the following equations, where $E_{Battmax}$ is the maximum energy stored in the battery, $E_{Batt}$ is the battery energy, and $P_{Batt}$ is considered to be the battery power. The $P_{DCBus}$ is considered to be the DC bus excess available power, and $\eta_{BDC/DC}$ is the DC/DC bidirectional-converter efficiency. In addition, $\eta_{dis}$ and $\eta_{ch}$ are considered to be battery discharge efficiency and battery charge efficiency, respectively. The energy of the battery is calculated at each moment, according to a recursive formula. This formula considers the amount of variation in the energy. The latter is calculated using the power available on the bus to be stored or consumed by the loads.

$$SoC(t) = \frac{E_{Batt}(t)}{E_{Battmax}} \tag{13}$$

$$E_{Batt}(t) = \int_{t-1}^{t} P_{Batt}(t) \times \Delta t \tag{14}$$

$$E_{Batt}(t) = E_{Batt}(t-1) + \Delta E_{Batt} \tag{15}$$

$$\Delta E_{Batt} = \begin{cases} P_{DCBus} \times \eta_{ch} \times \eta_{\frac{BDC}{DC}} & \text{Where, } \delta > 0 \\ \frac{P_{DCBus}}{\eta_{dis} \times \eta_{\frac{BDC}{DC}}} & \text{Where, } \delta < 0 \\ 0 & \text{Where, } \delta = 0 \end{cases} \tag{16}$$

$$SOC_{min} \leq SoC(t) \leq SOC_{max} \tag{17}$$

## 3. Methodology

### 3.1. Problem Formulation

In this paper, we study the utility of the combination of a quadratic sequential method with an artificial intelligence technique to solve the energy management issue of a hybrid renewable system. The system chosen for this study is the PV–wind–EV-battery connected to the grid. The hybrid system generally can be connected or not to the grid. Otherwise, the electric vehicle could be charge during trips or at the house during rests. This methodology influences how the energy produced by the hybrid system sources will be managed. Indeed, following the power estimation performed during the previous section, we propose a management system based on the linear programming of the energy flows communicated between each of the two actors within the architecture presented in Figures 1 and 2. These flows must satisfy the following balanced equation, according to each instant.

$$P_{PV2Bus} + P_{WT2Bus} + \delta P_{Batt} + \varepsilon P_{EV} - P_{Load} - P_{PV2grid} - P_{WT2grid} = 0 \tag{18}$$

where $P_{PV2Bus}$ is the PV energy flow produced to be consumed by household electrical loads and $P_{WT2Bus}$ is the wind turbine energy flow produced to be consumed by the same household electrical loads. In addition, $P_{PV2grid}$ and $P_{WT2grid}$ represent the energy to be injected as surplus to the neighboring micro-grid of the PV plant and the wind turbine, respectively. $P_{Batt}$ is the power made available by the battery and $\delta$ is the coefficient that reflects the battery state, as explained by the system (20). $P_{EV}$ is the contribution power of the electric vehicle and $\varepsilon$ is the flag that manages the status of the mobile battery represented by the EV. This flag could take three values, as represented in Equation (21). $P_{Load}$ is the total power demanded by the different electrical loads installed inside the household.

The coefficient must be negative when the battery accepts being charged with an availability of surplus energy produced by the hybrid system. Otherwise, the coefficient is positive, representing the discharge phase when the hybrid system is not able to fully satisfy the energy demanded by the house loads. The system where the battery must be at rest is represented by the phase where the state of charge is in its maximum state. This will be translated into the cancellation of the battery power in the optimization system; otherwise, the battery will not collaborate when its state of charge is minimum and the hybrid system not able to satisfy all the energy demanded by the house loads. For the electric vehicle, we check first the availability of the local battery before utilizing the electric vehicle mobile battery. The objective function of our optimization is to keep Equation (18) close to zero, so the objective is to minimize the norm of this equation:

$$\min_{P} F(P) = \min_{P} \left\| P_{PV2Bus} + P_{WT2Bus} + \delta P_{Batt} + \varepsilon P_{EV} - P_{Load} - P_{PV2grid} - P_{WT2grid} \right\| \Delta t \tag{19}$$

$P = \left[ P_{PV2Bus}, P_{WT2Bus}, P_{Batt}, P_{EV}, P_{PV2grid}, P_{WT2grid} \right]$ is the decision vector or all the system power flow that influences the balance of the energy ecosystem, where $P_{PV2Bus}$ and $P_{WT2Bus}$ are the shared power between the PV and the wind turbine with the DC bus, respectively. $P_{PV2grid}$ and $P_{WT2grid}$ are the power shared between the PV and the wind turbine with the neighborhood micro-grid, respectively. $P_{Batt}$ and $P_{EV}$ represent the local and mobile storage system, respectively.

Linear and nonlinear constraints are considered for proper functionalities of the system. In fact, each power flow from a hybrid source cannot exceed the maximum power that can be supplied by the system itself. The sum of the power supplied by the same source and

that will be consumed in two different flows also cannot exceed the maximum energy. This is implied by the solar system and the wind turbine. However, a renewable energy source cannot supply energy to the DC bus and to the grid at the same time. The storage system is dedicated to, or designed to last for, a period of 3 days, so we added a constraint on the power/energy that it can share over each hour to avoid premature degradation of the used batteries. These constraints are presented as follows.

$$\begin{cases} \delta = -1 \text{ if}(P_{PVMPP} + P_{WTMPP} \geq P_{Load} \&\&SoC(t) < SOC_{max}) \\ \delta = +1 \text{ if}(P_{PVMPP} + P_{WTMPP} \leq P_{Load} \&\&SoC(t) > SOC_{min}) \\ \delta = 0 \text{ if}(P_{PVMPP} + P_{WTMPP} \leq P_{Load} \&\&SoC(t) = SOC_{min}) || (P_{PVMPP} + P_{WTMPP} \geq P_{Load} \&\&SoC(t) = SOC_{max}) \end{cases} \tag{20}$$

$$\begin{cases} \varepsilon = -1 \text{ if}(P_{PVMPP} + P_{WTMPP} \geq P_{Load} \&\&SoC(t) = SOC_{max} \&\&SoCEV(t) < SOCEV_{max}) \\ \varepsilon = +1 \text{ if}(P_{PVMPP} + P_{WTMPP} \leq P_{Load} \&\&SoCEV(t) > SOCEV_{min} \&\&SoC(t) = SOC_{min}) \\ \varepsilon = 0 \text{ if}(P_{PVMPP} + P_{WTMPP} \leq P_{Load} \&\&SoCEV(t) = SOCEV_{min}) || (P_{PVMPP} + P_{WTMPP} \geq P_{Load} \&\&SoCEV(t) = SOCEV_{max}) \end{cases} \tag{21}$$

$$P_{PV2Bus} + P_{PV2grid} \leq P_{PVMPP} \tag{22}$$

$$P_{PV2Bus} \times P_{PV2grid} = 0 \tag{23}$$

$$P_{WT2Bus} + P_{WT2grid} \leq P_{WTMPP} \tag{24}$$

$$P_{WT2Bus} \times P_{WT2grid} = 0 \tag{25}$$

$$0 \leq E_{Batt} \leq E_{Battmax}(SOC_{max} - SOC(t)) \tag{26}$$

$$0 \leq E_{EV} \leq E_{EVmax}(SOCEV_{max} - SOCEV(t)) \tag{27}$$

As a result, the system that needs to be solved is represented as follows.

$$\min_P F(P) = \min_P \| P_{PV2Bus} + P_{WT2Bus} + \delta P_{Batt} + \varepsilon P_{EV} - P_{Load} - P_{PV2grid} - P_{WT2grid} \| \Delta t$$

$$\text{Where} \begin{cases} \delta = -1 \text{ if}(P_{PVMPP} + P_{WTMPP} \geq P_{Load} \&\&SoC(t) < SOC_{max}) \\ \delta = +1 \text{ if}(P_{PVMPP} + P_{WTMPP} \leq P_{Load} \&\&SoC(t) > SOC_{min}) \\ \delta = 0 \text{ if}(P_{PVMPP} + P_{WTMPP} \leq P_{Load} \&\&SoC(t) = SOC_{min}) || (P_{PVMPP} + P_{WTMPP} \geq P_{Load} \&\&SoC(t) = SOC_{max}) \end{cases}$$

$$\text{And} \begin{cases} \varepsilon = -1 \text{ if}(P_{PVMPP} + P_{WTMPP} \geq P_{Load} \&\&SoC(t) = SOC_{max} \&\&SoCEV(t) < SOCEV_{max}) \\ \varepsilon = +1 \text{ if}(P_{PVMPP} + P_{WTMPP} \leq P_{Load} \&\&SoCEV(t) > SOCEV_{min} \&\&SoC(t) = SOC_{min}) \\ \varepsilon = 0 \text{ if}(P_{PVMPP} + P_{WTMPP} \leq P_{Load} \&\&SoCEV(t) = SOCEV_{min}) || (P_{PVMPP} + P_{WTMPP} \geq P_{Load} \&\&SoCEV(t) = SOCEV_{max}) \end{cases} \tag{28}$$

$$\text{Subject to} \begin{cases} P_{PV2Bus} + P_{PV2grid} \leq P_{PVMPP} \\ P_{WT2Bus} + P_{WT2grid} \leq P_{WTMPP} \\ 0 \leq E_{Batt} \leq E_{Battmax}(SOC_{max} - SOC(t)) \\ 0 \leq E_{EV} \leq E_{EVmax}(SOCEV_{max} - SOCEV(t)) \\ P_{PV2Bus} \times P_{PV2grid} = 0 \\ P_{WT2Bus} \times P_{WT2grid} = 0 \end{cases}$$

The decision vector will represent a database of values representing the powers that will be shared with the bus or the grid. These values give an optimal combination as close as possible to the balanced equation presented in Equation (19). The system resolution will be conducted by sequential quadratic programming in MATLAB (23.2.0.2521687 (R2023b) Update 7). After the simulations, the management results will be used to propose an optimal hybrid system combination with LSTM prediction.

### 3.2. Sequential Quadratic Programming

By analyzing the proposed energy-management-system optimization strategy, we can deduce that the objective function is a minimization function that is linear. In addition, the constraints contain linear and non-linear terms. The constraints are performed with equalities and inequalities. This kind of problem formulation can be solved with sequential quadratic programming. This method has been first proposed by Wilson to solve constrained nonlinear optimization problems [40]. Afterwards, this method has gained interest in solving similar problems to overcome the difficulty of complex mathematical formulation [41]. To this end, we started by representing the energy management formulation in the canonical form, as is clear in system (29).

$$\min_{x \in \mathbb{R}^n} f(x)$$

$$\text{s.t.} \begin{cases} C(x) \le 0 \\ C_{eq}(x) = 0 \\ A.x \le b \\ A_{eq}(x) = b_{eq} \\ l_b \le x \le u_b \end{cases} \tag{29}$$

where $f : \mathbb{R}^n \to \mathbb{R}^n, C : \mathbb{R}^n \to \mathbb{R}^m, C_{eq} : \mathbb{R}^n \to \mathbb{R}^p, A : \mathbb{R}^n \to \mathbb{R}^q$ and $A_{eq} : \mathbb{R}^n \to \mathbb{R}^t$. A and $A_{eq}$ are functions representing linear inequalities and equalities, respectively. C and $C_{eq}$ are functions summarizing the non-linearity of the system constraints in its inequalities and equalities forms, respectively. Moreover, the $L_b$ and $U_b$ represent the vectors that contain the upper and lower limits of the x vectors. To solve this problem, we need to represent the system in its matrix form, as shown by Equations (30)–(33).

Absent parameters in the function formulation are translated by an empty matrix or vector, which is the case for C and $A_{eq}$ and $b_{eq}$. For our simulation, we need to express this formulation for each time slot, which means for one hour. Otherwise, the simulation could be conducted for each time step that characterized the meteorological condition dataset.

$$x = \begin{bmatrix} P_{PV2Bus}, P_{WT2Bus}, P_{Batt}, P_{EV}, P_{PV2grid}, P_{WT2grid} \end{bmatrix} \tag{30}$$

$$C = [\ ]; C_{eq} = \begin{bmatrix} P_{PV2Bus} \times P_{PV2grid} \\ P_{WT2Bus} \times P_{WT2grid} \end{bmatrix} \tag{31}$$

$$A_{eq} = [\ ]; b_{eq} = [\ ]; A = \begin{bmatrix} 1\ 0\ 0\ 0\ 1\ 0 \\ 0\ 1\ 0\ 0\ 0\ 1 \end{bmatrix}; b = \begin{bmatrix} P_{PVMPP} \\ P_{WTMPP} \end{bmatrix} \tag{32}$$

$$l_b = [0\ 0\ 0\ 0\ 0\ 0], u_b = [P_{PVMPP}\ P_{WTMPP}\ P_{Battmax}\ P_{EVmax}\ P_{PVMPP}\ P_{WTMPP}] \tag{33}$$

Every parabola representation of a function has a vertex. This latter could represent a maximum or a minimum value of the function. This function represents the objective function that we need to satisfy. This definition is expressed by quadratic optimization [42]. Indeed, sequential quadratic programming is a technique that generates iterates that are trying to convert to an optimal solution of the formulated problem (ex. Equation (30)), by solving the quadratic programs [43]. The idea behind quadratic sequential programming is to approach the problem we are trying to solve by means of quadratic sub-problems, according to each iteration, so that in the end they converge towards the global optimal solution. The iterative solution is checked each time and improved in a direction defined by the quadratic sub-problem, which ensures that all the constraints are satisfied. The system stops at a point where the following iterations do not improve the solution in a meaningful way. For more details, please refer to Chapter [44], Subsection 1.16.6.2.3.

### 3.3. LSTM Prediction

Our dataset represents the energy collaboration of every source of the hybrid system. Indeed, over a time step of the simulation the household must consume an amount of energy. This electrical demand needs to be first ensured by the hybrid renewable system, the storage system, the electrical vehicle collaboration, and then the utility grid. The storage system can be a load or a source, depending on the potential renewable availability. Therefore, the proposed energy management strategy will give the optimal energy to be produced, consumed, or stored in every single simulation time step. This is categorized under the nomination of time series, representing our data proposed over a regular period. To process and predict these kind of data, time-series algorithms are needed. In this work, we will use a long short-term memory (LSTM) algorithm. LSTM is an advanced type of recurrent neural network that can process sequences of data and, likewise, time series. As a

result, it can predict these types of data based on previous values, successfully [45]. The most well-known LSTM variant is described by equations, represented as follows [46].

$$f_t = \sigma \times (W_{fh} \times h_{t-1} + W_{fx} \times x_t + b_f) \tag{34}$$

$$i_t = \sigma \times (W_{ih} \times h_{t-1} + W_{ix} \times x_t + b_i) \tag{35}$$

$$o_t = \sigma \times (W_{oh} \times h_{t-1} + W_{ox} \times x_t + b_o) \tag{36}$$

$$\widetilde{c}_t = \tanh(W_{ch} \times h_{t-1} + W_{cx} \times x_t + b_c) \tag{37}$$

$$c_t = f_t \odot c_{t-1} + i_t \odot \widetilde{c}_t \tag{38}$$

$$h_t = o_t \odot \tanh(c_t) \tag{39}$$

where $f_t$ represents the forget gate, $i_t$ is the input gate, $o_t$ is the output gate and $c_t$ is the memory cell. $f_t$, $i_t$, and $o_t$ represent sigmoid layers. The system follows a process to eliminate error reproduction from the previous prediction to the subsequent one. The process starts by deciding the significant information that we will keep for the next prediction and the one that we need to eliminate to avoid potential errors: this process is described by Equation (34). In fact, a value near to 0 needs to be eliminated and values near to 1 should be kept. The following step is to create a new candidate in $\widetilde{c}_t$ following Equation (37). After that, the input gate sigmoid, Equation (35), will decide which values should be stored in the memory cell $c_t$, Equation (38), depending on the previous memory-cell values. Finally, based on the memory cell and the output gate $o_t$, the system will decide the output values that should be kept using the tanh, which converts the memory-cell numbers into the interval of $[-1, 1]$ [47].

### 3.4. Random Forest

This is a method that was first developed by Breiman [48], which is adopted for prediction and classification. The idea is to split the data into multiple samples; next, we construct a random tree for each group of sub-data, then average the obtained prediction in each segment. The random forest process follows essential steps each time the system is used for prediction. First we have a training database that is independent and has the form of a pair (x, y), where x is the input features and y is the target to be predicted. The random forest procedure starts by taking random values from the original database and building small databases of the same size each time, i.e., N sub-databases [49,50]. Each tree can give a prediction of a point $x_{new}$ which is currently untrained the first time, using the following function:

$$y_{new,j} = f_j(x_{new}) \tag{40}$$

where $y_{new,j}$ is the predicted value of the $x_{new}$ using the j-th decision tree by its function process $f_j$. After calculating the estimation of each decision tree's prediction, an averaging process is considered, according to the following formulation:

$$y_{new} = \frac{1}{N} \sum_{i=1}^{N} f_j(x_{new}) \tag{41}$$

where $y_{new}$ is the final prediction that is estimated by using the random forest method.

### 3.5. Gradient Boost Machine

When a complex non-linear relationship exists between the input features and the output target, there is no solution better than using the gradient boost machine [51]. This is a machine learning algorithm that combines prediction from weak learners; generally, its predictions are made by decision trees, as is the case for random forest. The algorithm start by supposing that we have a dataset of N size, having the form of $(x_i, y_i)$, where $x_i$ represents the features and $y_i$ is the corresponding target, and, obviously, the I can take

a value from 1 to N. We suppose, also, that we then want to have a prediction of the following target:

$$\hat{y}_i = F(x) \tag{42}$$

The objective is to find an estimator that is capable of minimizing the mean square error of the training set, defined as follows:

$$\text{MSE} = \frac{1}{N}\sum_{i=1}^{N}(y_i - \hat{y}_i)^2 \tag{43}$$

We consider that for a model noted as $F_g$ that fits with accuracy the training set, the process of the gradient boost will add a new estimator hg, which will then satisfy the following equation:

$$y = F_g(x) + h_g(x) \tag{44}$$

This means that the estimator hg is fitting the errors made by the previous model $F_g$. Through calculation, the gradient of the loss function L of the previous model is negatively proportional to the new estimator that fits the errors made by the $F_g$ [52,53].

$$\frac{\partial L}{\partial F} = \frac{\partial\left(\frac{1}{N}(y - F(x))^2\right)}{\partial F} = -\frac{2}{N}(y - F(x)) = -\frac{2}{N}h_g(x) \tag{45}$$

The goal, then, of the gradient boosting is to add in every iteration an estimator that tries to correct errors made by the previous model by minimizing the loss function.

*3.6. KNN*

This algorithm is considered as the simplest algorithm among machine learning algorithms, whether for regression or classification [54]. Let us consider that we have a dataset defined in the following description:

$$D = \{(X_i, y_i)\}_{i=1}^{N}, X_i \in \mathbb{R}^n \tag{46}$$

where N is the size of the dataset, $X_i$ represents the different features, and $y_i$ is the corresponding output target for a new $X_{new}$ value that represents a new value that was not considered in the first dataset D. The algorithm started by determining the k value, which represents the number of neighbors that the system needs for an efficient prediction [55]. Then the system calculates the distance between the new point searched and all points in the dataset, following one of the equations described by (47)–(49), representing Euclidean distance, Manhattan distance, and Minkowski distance, respectively [56]. We should mention that the Minkowski distance is a generalized form of the Manhattan and Euclidean distances, achieved by replacing p with 1 and 2, respectively.

$$d = \sqrt{\sum_{i=1}^{n}(x_i - y_i)^2} \tag{47}$$

$$d = \sum_{i=1}^{n}(x_i - y_i) \tag{48}$$

$$d = \sqrt[p]{\sum_{i=1}^{n}(x_i - y_i)^p} \tag{49}$$

The following step is to compare all the calculated distances and keep just the closest k neighbors and use their labels to predict the target label by averaging or weighting all the selected labels [57,58]. Then the prediction is made by the following equation:

$$\hat{y}_{new} = \frac{1}{k}\sum_{i=1}^{k} y_i \tag{50}$$

## 4. Results and Discussion

The hybrid renewable energy system (HRES) architecture is composed of a solar system and a wind turbine system, with battery storage and electric vehicle all connected to the grid, as clarified in Figures 1 and 2. To validate the efficiency of our prediction procedure, this HRES is used to ensure the demand for loads installed inside a house located in the city of Marrakech in Morocco, with coordinates 31°37′48″ N, 8°00′00″ W. The energy required by the house is estimated at 12 KWh/day for a residential building housing four people, which is the average size of a Moroccan household according to studies initiated by the High Commissariat of the Plan (HCP) [59]. The starting simulation was carried out with the values summarized in Tables 1–3, representing all the characteristics of the wind turbine and the solar panel and electric vehicle.

**Table 1.** Wind turbine parameters.

| Parameter | Value |
|---|---|
| Rated turbine value | 2.4 kW |
| Rotor radius | 1.86 m |
| Air density | 1.225 kg/m$^3$ |
| Number of pole pairs | 8 |

**Table 2.** Photovoltaic panel parameters.

| Parameter | Value |
|---|---|
| Maximum power | 300 Wp |
| Short-circuit current | 9.06 A |
| Open-circuit voltage | 44.52 V |
| Temperature coefficient of $V_{co}$ | −0.346%/°C |
| Temperature coefficient of $I_{sc}$ | +0.036%/°C |
| Number of cells | 72 |
| Ideality factor | 1.5 |

**Table 3.** Electric vehicle parameters.

| Type | Electrical Power |
|---|---|
| Ebatt | 5.5 kWh |
| Autonomy (Max.) | 70 Km |
| Speed (Max.) | 45 Km/h |
| Charging | 3 h |

The proposed energy management is based on linear optimization with non-linear constraints. Indeed, the objective is to find the right collaboration between the renewable sources (solar, wind), the storage system, and the electric vehicle, without forgetting that in the case of deficit the system will resort to the grid. The objective function that the system must satisfy for each time is to minimize the grid consumption. As a result, the system uses the grid just during critical periods when we have no renewable potential available and the storage system is at its minimum state of charge. Fluctuations in renewable sources are caused by the availability or lack of renewable potential. A modeling of the renewable

sources has been made at the level of Equations (1)–(8), making the renewable system production a mathematical formula which estimates the production of the renewable sources according to their renewable potential. This amount of energy must match the electricity consumption of the building, which is constituted in its turn of several loads: their total consumption is the object of the power $P_{Load}$ that we seek to satisfy, by means of the renewable sources, the battery storage, the presence of the electric vehicle, and the presence of the grid. This satisfaction is the aim of the minimization objective set out in Equation (17). To test the validity of our proposal, we considered an algorithm executed on MATLAB. Indeed, Algorithm 1 represents the pseudo-algorithm followed for the implementation of our management strategy on MATLAB.

---

**Algorithm 1:** Energy-management-system linear programming

---

**Result**: Optimal Energy Collaboration
*Initialization*:
*Extract meteorological conditions G, Vwind, T;*
*Collect SoC(t), $SoC_{max}$, $SoC_{min}$, $E_{Battmax}$, $E_{EVBatt}$, SoCEV(t), $SoCEV_{max}$,*
*$SoCEV_{min}$, $E_{EVBattmax}$, $P_{Load}$;*
*Initialize SoC, SoCEV;*
*Define P as [P(1) P(2) P(3) P(4) P(5) P(6)]*

**While** $P_{Load} \neq 0$ **do**
  Calculate $P_{PVMPP}$, $P_{WTMPP}$;
  $E_{Batt} = E_{Battmax}(SoC_{max} - SoC(t))$ ;
  $E_{EVBatt} = E_{EVBattmax}(SoCEV_{max} - SoCEV(t))$ ;
  Initialize linear programming parameters (C, $C_{eq}$, A, $A_{eq}$, b, $b_{eq}$, $U_b$, $L_b$)using the Canonical form
  Determine the coefficient value δ for battery storage system application.
  Determine the coefficient value ε for electric vehicle collaboration.
  Define fun as $||P(1) + P(2) + P(3) + P(4) - P_{Load} - P(5) - P(6)||$;
  Minilise objective function
  Update SOC(t);
  Update SOCEV(t);
**end**

---

For the simulation we considered four typical days, one day for each season of the year. Figure 3 represents the simulations retained. Results show that the program tries to find the most optimal production combination between the energy consumption that could be satisfied by wind production, solar production, or by the combination of the two. Besides that, there is the presence of the grid and the storage system using batteries that also charge the electric vehicle. These are taken into consideration during a renewable-energy deficit. We can deduce that the proposed system manages the energy whatever the season of the year it is. We notice that our proposed system manages the energy from the hybrid system, regardless of the day of the year, and in any season. The system management proposes that the household consumes energy from the hybrid system with a certain percentage assigned to each renewable source. Once the hybrid system is no longer able to meet the demand or generates more energy than is required, the storage system and electric vehicle come into operation. In the case of a power surplus and when the storage system and/or electric vehicle is below the limit of its state-of-charge value, the charging mode is activated. In the opposite case, the discharge mode is activated to meet the load, until the electrical vehicle and storage system reach their minimum state of charge. Depending on the day, we notice that generally the loads of the house consume hybrid energy, between solar and wind production.

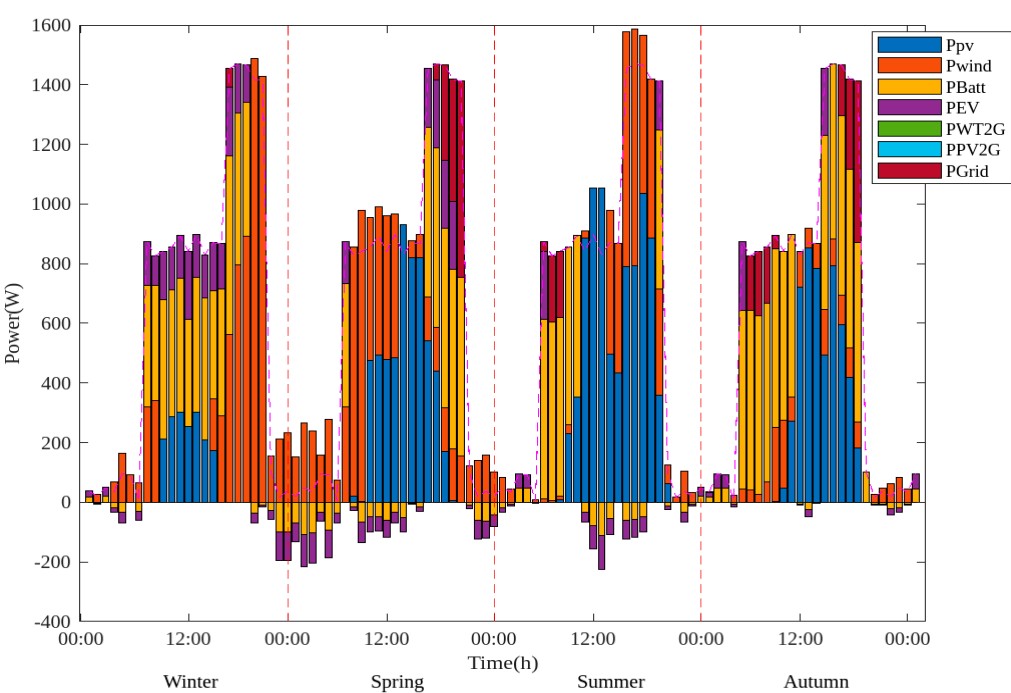

**Figure 3.** Energy-management-system MATLAB simulation.

However, there are seasons when the solar system takes over and covers the entire electrical demand of the house. This is the case for the summer and autumn seasons, around 12:00. This phenomenon varies from one implementation site to another, as renewable generation is intermittent and highly correlated to weather conditions. Furthermore, in other seasons we have periods where the wind source is the only source that satisfies the demand; this is clear during the spring in the morning or after sunset during the winter and spring. In other periods, the system uses the storage system or the electric vehicle relies on the grid. We also observe from the results that, occasionally, the production of the hybrid systems exceeds the demanded consumption. This case is related to the presence of negative storage-system powers, and represents the periods found by the optimal management system for local and mobile-battery charging. This management was then generalized for a one-year simulation. The obtained results for a year were then processed using LSTM to predict the production of each source during renewable energy consumption.

By analyzing the resulting energy management database in MATLAB, we can deduce that the contribution of each source is different from zero throughout all the simulation steps. The management system favors the use of renewable energies instead of conventional fossil fuels supplied to the national grid by thermal power stations. However, it does not promote grid injection. Consequently, the contribution of local sources to neighborhood demand is virtually non-existent. Instead, the storage system is provided by the battery system and the residents' electric vehicles. The $P_{PV}$ and $P_{WT}$ represent the use of solar and wind energy, respectively. These two sources are intermittent and follow the night/day and seasonal changes in the year. This implies that this database is a time series, and is influenced by the renewable potential that is controlling it. To achieve this, we set out to predict the $P_{PV}$ and $P_{WT}$ using the G, T, $V_{wind}$ and $P_{Load}$ representing solar irradiation, ambient temperature, wind speed, and the power demanded by the building, respectively. The prediction was carried out with the LSTM model. Below, in Figure 4, is the approach used for the prediction. It is a four-input (input layer), two-output model (output layer) with a first LSTM layer and three dense layers. The nodes on each layer have been set to 512, except the first dense layer, which was performed with 1024. Initially, the database was split in two, with 30% for test and validation as well as 70% kept for training. The learning database was divided during model development into two sub-datasets, with a percentage of 20% for testing and 80% for understanding and learning.

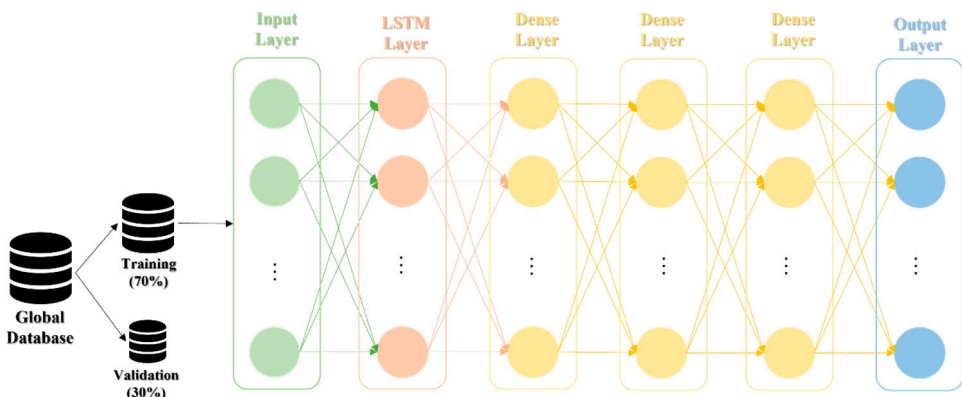

**Figure 4.** LSTM-regressor adopted approach.

As a result, the model gave us an accuracy of 94% during training, with an RMSE and MAE of 69.40 and 29.63, respectively. This implies that the predicted energy may have been made with an error of about 30 W. In our case, where the energy we are looking for is around 13 kW, the 30 W is insignificant. These parameters were noticed as even smaller for the test and validation database. In fact, the R2_score was around 95%. This implies that the prediction system developed could forecast the desired data with a precision up to 95%. The system was tested with a non-normalized database, but the prediction fell in accuracy compared with the normalized version. The results of LSTM are shown in Figures 5 and 6, which show the wind turbine and PV production predictions, respectively. Indeed, Table 4 was established according to the different types of normalization and data regularization. When we tested the ability of the system to predict an output of parameters at the same time, namely, $P_{PV}$, $P_{WT}$, $P_{batt}$ and $P_{EV}$ at once, the system returned an accuracy that was slowed down to 64%. This prompted the study to consider hybridizing this model with another machine learning algorithm, which may potentially be beneficial. In our case, because the two remaining parameters are highly fluctuating and have negative values, we tested the combination with the random forest, gradient boost, and k-nearest neighbors algorithms. Furthermore, the LSTM did not give a significant accuracy for this specific prediction. Therefore, the solution was to take both outputs of the LSTM system and predict the $P_{EV}$ and $P_{Batt}$ using the above-mentioned algorithms, since the values that influence the storage system are obviously the availability of energy in its renewable or conventional forms and, imperatively, the demand for energy. Consequently, the comparison is summarized in Table 5.

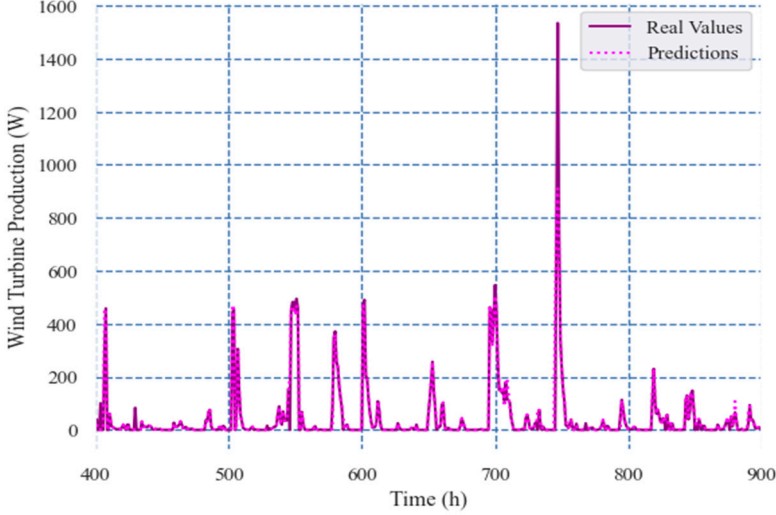

**Figure 5.** LSTM-regressor prediction for wind turbine power.

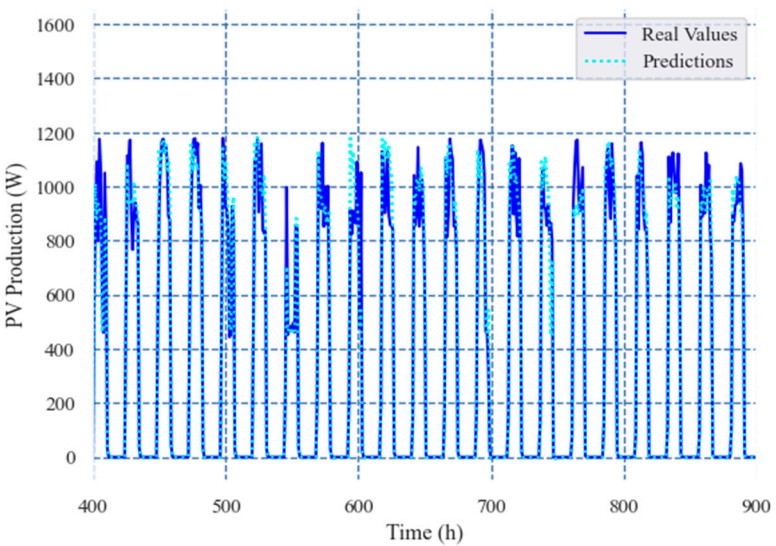

**Figure 6.** LSTM-regressor prediction for PV power production.

**Table 4.** Data-normalization-method comparison.

| Model | Data Normalization | RMSE | MAE | R2_Score |
|---|---|---|---|---|
| | None | 67.0591 | 25.0886 | 0.9533 |
| | Min–Max scaling | 72.7265 | 32.27 | 0.9418 |
| | Z-score scaling | 70.7165 | 30.3589 | 0.9404 |
| | Log transformation | Nan | Nan | 0.5292 |
| | Quantile transformation | 129.6644 | 56.8839 | 0.6308 |
| | Robust scaling | 148.9046 | 65.9128 | 0.4322 |
| Long Short-Term | Feature scaling to specific range | 68.3868 | 30.6786 | 0.9468 |
| Memory network | Decimal scaling (3) | 146.4857 | 66.1624 | 0.4400 |
| | Max absolute scaling | 71.2652 | 31.5538 | 0.9425 |
| | Softmax transformation | 469.0968 | 238.6566 | −0.3430 |
| | Power transformation (2) | 171.3721 | 69.1538 | 0.7624 |
| | Unit vector scaling | 163.5302 | 76.9106 | 0.3916 |
| | Max norm scaling | 163.0177 | 73.1468 | 0.4215 |

**Table 5.** Tested machine learning algorithm comparison combined with LSTM output layers.

| Model | RMSE | MAE | R2_Score |
|---|---|---|---|
| Random Forest | 74.6369 | 50.3825 | 0.6953 |
| Gradient boosting | 87.7846 | 57.8135 | 0.7391 |
| K-nearest neighbors | 56.5445 | 35.3834 | 0.8381 |

In fact, in our case we take the results of the LSTM model and put them as inputs for each chosen machine learning algorithm, beginning with the algorithm of random forest, which chooses randomly columns from the input and uses them to predict the target value. The way of prediction is different from one tree to another. The final prediction is then made by voting or averaging all predictions made by the *n* trees of the algorithm. This process is explained in the following Figure 7.

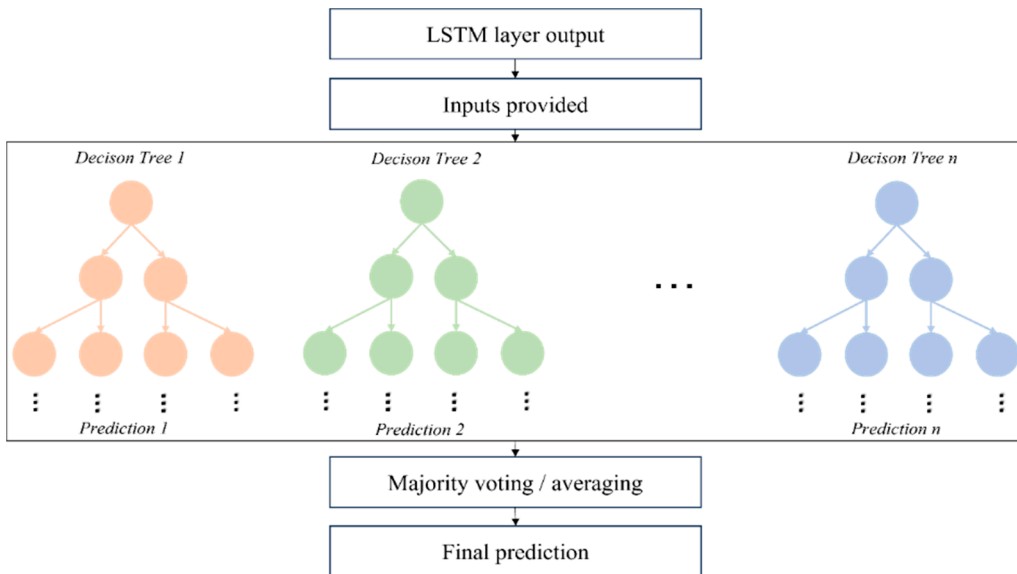

**Figure 7.** Random forest process.

In the case of KNN, the process in our paper begins by tracking the output values obtained from the LSTM for renewable system collaboration and the energy demanded by the household. After that, the algorithm is characterized by the k value, which represents the number of neighbors that the system needs for an efficient prediction. Then the system calculates the distance between the new point searched and all points in the dataset, by using one of the following equations that represents the Euclidean distance, the Manhattan distance, and the Minkowski distance, respectively. The following step is to compare all the calculated distances and keep just the closest *k* neighbors and use their labels to predict the target label by averaging or weighting all the selected labels.

In our paper we also tested the performance of the gradient boost algorithm to predict values from LSTM layer inputs. We used a sequential process against the parallelism execution followed by the random forest algorithm. Generally, the trees used in gradient boost are one-depth trees, with one node source and two child nodes. The *t* + 1 tree used tries to correct the error caused by the *t* tree, and vice versa. So, to predict the value of a new point, the system needs to run a prediction from all the trees, and from one tree to another the prediction become more accurate. The process adopted in this paper is presented in Figure 8.

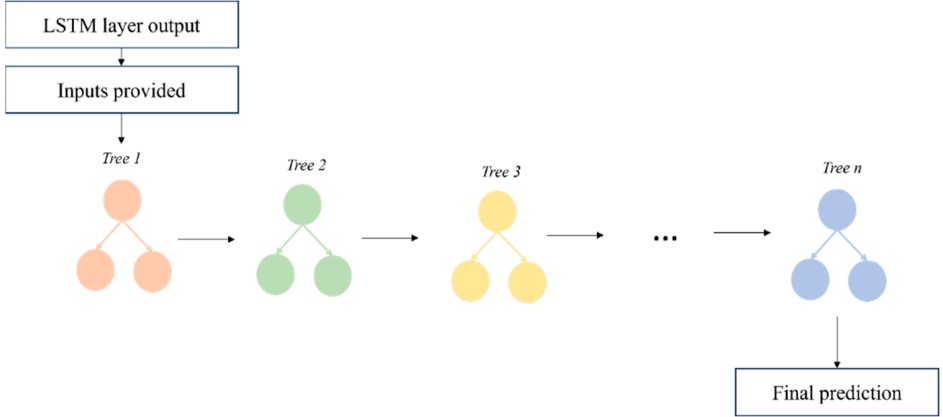

**Figure 8.** Gradient-boosting-regressor process.

The random forest, in its turn, gave us an R2-score of 0.6953 with the same order of RMSE and MAE. Otherwise, the KNN gave the best R2_score of 0.8381, with ameliorated

MAE and RMSE values. Data normalization gave no advantage in terms of accuracy or error for these algorithms. In addition, the data required cleaning and pre-processing before training. This model provided accuracy at the level of rapid and abrupt fluctuation. This is clear from Figures 9 and 10 below. They show the prediction of battery storage and electric vehicle collaboration using the KNN algorithms.

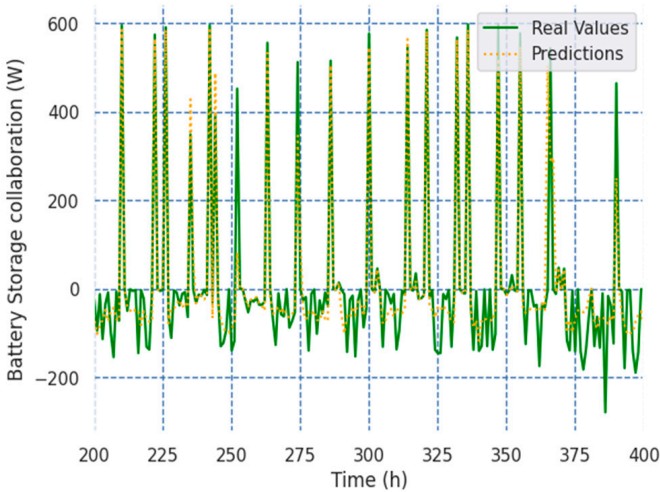

**Figure 9.** KNN-regressor prediction for battery storage collaboration.

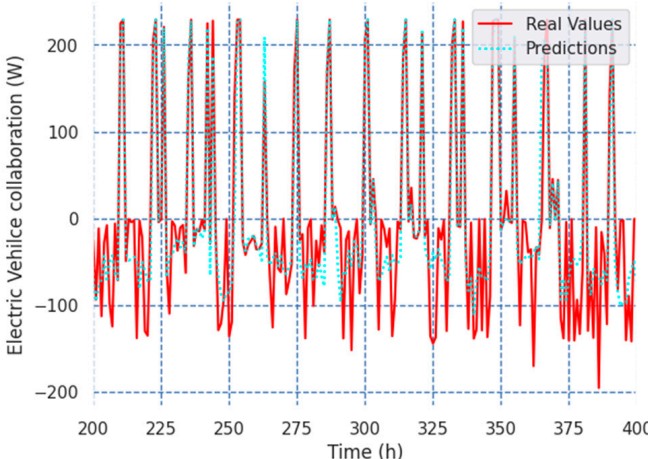

**Figure 10.** KNN-regressor prediction for electric vehicle collaboration.

Prediction using LSTM-KNN hybridization appears to be the most accurate method. The purpose of this model is to predict the energy that will be produced by renewable sources and by the storage system, as well as the collaboration of the electric vehicle. This will obviously allow us to predict in advance the control and switching states of the switches mentioned in Figures 1 and 2 above, with the amount of energy produced by the system and available for consumption by the loads installed in the building concerned. These results have been the subject of a third LSTM model that will even allow us to predict the amount of energy that will be consumed from the national grid and, consequently, the monthly energy bill for each building. The obtained results that are presented in Figure 11 represent a prediction of an R-square ($R^2$) of about 0.9834 and a mean absolute error of 4.5112. These values mean that our system can predict the energy that could be consumed from the grid with an error of less than 5 W, which help us to predict the energy that could be consumed at the end of the month and the bills to pay, with a high level of accuracy. In our study, the combination of deep learning and machine learning methods ultimately produced a prediction with a high R2_score. In fact, the final R2_score can be considered initially as an

example of the precision of the set, given that the result which is the consumption from the utility grid was compared with the previously defined management system results. In fact, the model is a combination of LSTM and KNN, but each prediction part treats a variable to be predicted separately, which constitutes a feature that influences the following prediction block.

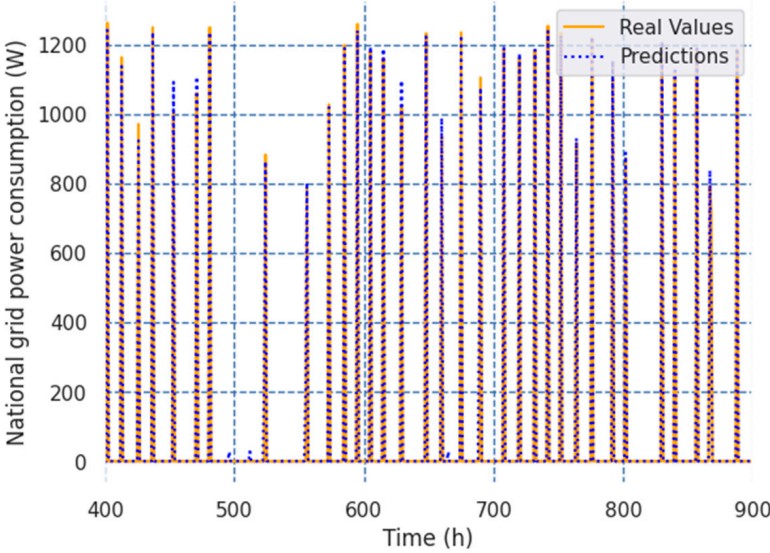

**Figure 11.** LSTM-regressor prediction for national-utility-grid power consumption.

Our dataset considers the most important aspect, which is the notion of time. So, the data is a time series in nature, making it necessary to focus on the use of algorithms that process these types of data. One of these algorithms, the LSTM used here, is distinguished by its reputation for predicting these types of data with a high level of accuracy. That is why it has performed well for photovoltaic and wind-power systems. However, for other items of data we need to predict, such as battery capacity and electric vehicle power, even though they are correlated with time, their fluctuating nature and repetitive values due to charging and discharging limitations require a data pre-processing approach and a machine learning algorithm such as KNN, which is less sensitive to fluctuations, since the output is estimated using neighbor points following a predefined distance.

The proposed system has been tested and validated based on simulations carried out in Marrakech, which can be generalized to several locations. The proposed hybrid model is based on the PV–wind–battery–EV system, and its modeling considers the meteorological conditions and the presence or absence of the electric vehicle. This implies that all the derivatives of this hybrid system are valid for consideration by our optimal management system using the quadratic optimization model. Furthermore, the fact that it is linked to meteorological conditions makes it possible to estimate the output of the hybrid system. Moreover, our estimate of electricity consumption inside the house is ensured by means of a load power balance, and hence the size of the building in no way affects the performance of our proposed system.

The management system is designed to minimize, in the positive sense of the term, a balanced equation, following constraints that ensure a safe use of the hybrid system. The idea behind this minimization is to ensure a balance between what is produced and what is consumed by the household loads. This ensures the monitoring of the availability of renewable production. Indeed, when the system is in the surplus period, the energy is stored in the battery storage system or in the electric vehicle battery; otherwise, it is injected into the grid. Any intermittency on the part of renewable sources is remedied by the presence of the battery storage system and/or the electric car battery. Alternatively, our system can turn to the grid to meet any electricity demand that has not been met by the renewable or storage system. The proposed management system in this paper takes into

consideration the use of a battery storage system and the electric vehicle and, moreover, the presence of the utility grid as a backup system during critical cases. The nonlinear constraints ensure the energy balance between the produced and consumed energy at each time step of the algorithm. As a result, the amount of energy supplied by the grid is located during the night and before sunrise, which means that the time of use of the electricity hardly influences our approach.

The approach concerns a minimization objective function subject to linear and non-linear constraints. Linear constraints are generally easy to deal with; however, non-linear constraints increase the complexity of solving the program that is optimized with the use of sequential quadratic programming. Otherwise, the number of decision variables is six, which is a small number of variables resulting in a lower computational complexity. The solution method used in our case is the interior-point method, which can approach the solution iteratively. The computational complexity of this approach is typically around $O(n^2)$ and $O(n^3)$, where n is the number of decision variables.

The implementation of our system is subject to several parameters that we need to consider. Initially, the availability of renewable potential is required to ensure the possibility of a renewable system based on PV–wind–battery–EV or its derivatives. Secondly, we need to adopt a set of connected sensors to collect data on the climate and/or the production of renewable sources, as well as the consumption of electrical loads inside the house. This gives rise to the notion of the communication protocol and the connection between this set of sensors and the data processing center. Furthermore, a study will be made of the legislation governing the implementation of small-scale hybrid renewable production systems in each territory.

## 5. Conclusions

The integration of renewable energies has begun to be a worldwide trend. However, their use has given rise to problems of intermittency, which have been remedied by the hybridization of complementary renewable sources. This, in turn, has created problems in terms of energy management. The latter has been addressed in several ways, beginning with rule-based methods, through heuristic methods, to mathematical optimization programming. To this end, we have contributed to the development of a management system based on quadratic programming. Its solution was processed with the LSTM and the random forest machine learning algorithm to facilitate the treatment of possible cases without having to solve a complex system every time. The results gave accuracies of around 95% and 83% using the LSTM and KNN algorithm, respectively. In fact, the model was taken to be a sequential chain, which was fed by the database of results from the system solved using MATLAB, processed in LSTM and then in the KNN algorithm. Afterward, an LSTM model was used to forecast the energy that can probably be consumed from the national utility grid. This model took five features, namely, the two outputs of the first LSTM model ($P_{PV}$ and $P_{WT}$) and the two targets of the KNN model ($P_{Batt}$ and $P_{EV}$) and the load power consumption, which is also a decisive feature of this model. The final target was the $P_{grid}$, which represents the amount of power that could be consumed from the grid. The prediction was characterized by an R-score of 0.9834 and a mean absolute error of around 5 W.

**Author Contributions:** Conceptualization, A.C.; methodology, A.C.; software, A.C.; validation, A.C. and M.T.; formal analysis, A.C. and M.T.; investigation A.C. and M.T.; resources A.C and M.T.; data curation A.C. and M.T.; writing—original draft preparation, A.C.; writing—review and editing, A.C. and M.T.; visualization, M.T.; supervision, M.T.; project administration, A.C. and M.T.; funding acquisition, M.T. All authors have read and agreed to the published version of the manuscript.

**Funding:** This research received no external funding.

**Institutional Review Board Statement:** Not applicable.

**Informed Consent Statement:** Not applicable.

**Data Availability Statement:** The data that support the findings of this study are available from the corresponding author upon reasonable request.

**Acknowledgments:** The authors are thankful to referees for the constructive comments that contributed to the first draft of the paper's improvement. The authors are very grateful to the LPRI laboratory, EMSI Casablanca.

**Conflicts of Interest:** The authors declare no conflicts of interest.

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
