# Peer review of "Hybrid Renewable Production Scheduling for a PV–Wind-EV-Battery Architecture Using Sequential Quadratic Programming and Long Short-Term Memory–K-Nearest Neighbors Learning for Smart Buildings"

_sustainability, doi:10.3390/su16052218_

Round 1

Reviewer 1 Report

Comments and Suggestions for Authors

The paper presents a hybrid renewable energy system for residential buildings, integrating photovoltaic (PV), wind turbine, electric vehicle (EV), and battery storage. An energy management system based on nonlinear programming is proposed, solved using sequential quadratic programming. Data processing involves Long short-term memory (LSTM) combined with K-nearest neighbors (KNN) to predict the contribution of each energy source and storage element. Results indicate high prediction accuracy, with an R²_score of around 0.9834 for forecasting potential grid consumption using LSTM. The methodology offers insights into energy usage and facilitates efficient switching control for the hybrid architecture.

My Comments:

1) Can you elaborate on the criteria used to determine the optimal size of the hybrid renewable energy system for residential buildings?

2) How does the energy management system handle dynamic changes in renewable energy production and building demand to ensure efficient utilization of resources?

3) What specific challenges were encountered in integrating the electric vehicle charging terminal into the hybrid system, and how were they addressed?

4) Could you provide more details on the sequential quadratic programming approach used for solving the energy optimization problem?

5) How was the accuracy of the LSTM-KNN model assessed, and were there any limitations or uncertainties in the prediction results?

6) What considerations were made regarding the scalability of the proposed methodology for larger residential complexes or different geographical locations?

7) Can you discuss the computational complexity of the proposed energy management system and any strategies employed to optimize computational efficiency?

8) Were there any constraints or trade-offs considered in the selection of the LSTM-KNN approach for data processing and prediction?

9) How does the proposed methodology handle uncertainties in renewable energy generation, such as variations in weather conditions?

10) In real-world applications, what are the key factors influencing the practical implementation and adoption of the proposed hybrid renewable energy system in residential buildings?

Author Response

 Original Manuscript ID: sustainability-2845953

Original Article Title: Hybrid renewable production scheduling for a PV-Wind-EV-Battery architecture using sequential quadratic programming and LSTM-KNN learning for smart buildings

To: MDPI Sustainability

Re: Response to reviewers

Dear Editor, Dear Reviewer,

We appreciate the chance to revise and resubmit our work to address the concerns provided by the reviewers.

We have uploaded our detailed response to the remarks (the response to reviewers), along with a revised manuscript that includes highlighted revisions in yellow.

Best regards,

Asmae Chakir and Mohamed Tabaa

-------------------------------------------

The paper presents a hybrid renewable energy system for residential buildings, integrating photovoltaic (PV), wind turbine, electric vehicle (EV), and battery storage. An energy management system based on nonlinear programming is proposed, solved using sequential quadratic programming. Data processing involves Long short-term memory (LSTM) combined with K-nearest neighbors (KNN) to predict the contribution of each energy source and storage element. Results indicate high prediction accuracy, with an R²_score of around 0.9834 for forecasting potential grid consumption using LSTM. The methodology offers insights into energy usage and facilitates efficient switching control for the hybrid architecture. My Comments:

  • Can you elaborate on the criteria used to determine the optimal size of the hybrid renewable energy system for residential buildings?

We would like to acknowledge and express our thanks for all pertinent and constructive remarks. The size of the system was based on conventional sizing approach that was mentioned as demanded in introduction. Please refer to the lines [123-126].

  • How does the energy management system handle dynamic changes in renewable energy production and building demand to ensure efficient utilization of resources?

We thank you for this pertinent remark. We clarified even more this point by adding a paragraph in Results and discussion section. Please refer to the lines [467-476].

  • What specific challenges were encountered in integrating the electric vehicle charging terminal into the hybrid system, and how were they addressed?

The most challenges encountered in this integration are communication and data exchange, compatibility issues and power management. We highlighted these challenges in a paragraph in section 4.3. Please refer to the lines [235-244].

  • Could you provide more details on the sequential quadratic programming approach used for solving the energy optimization problem?

We appreciate this remark. We believe also that the description was not clear. We added more details in the concerned section. Please refer to the lines [341-348].

  • How was the accuracy of the LSTM-KNN model assessed, and were there any limitations or uncertainties in the prediction results?

In our case study, the combination of deep learning and machine learning methods ultimately produced a prediction with a high R2_score. We could base on the accuracy of the final target to evaluate the process in all. To clarify this point of view we added more details in the discussion section. Please refer to the lines [628-635].

  • What considerations were made regarding the scalability of the proposed methodology for larger residential complexes or different geographical locations?

The scalability of the proposed methodology depends on the potential availability and the infrastructure of the building. We add more explanation in the results and discussion section.  Please refer to the lines [646-655].

  • Can you discuss the computational complexity of the proposed energy management system and any strategies employed to optimize computational efficiency?

We appreciate your pertinent remark according to the complexity study of the proposed algorithm. We discussed that in the results and discussion section also. Please refer to the lines [672-676].

  • Were there any constraints or trade-offs considered in the selection of the LSTM-KNN approach for data processing and prediction?

Our dataset is timeseries. As a result, the LSTM was the best choice. However, the data concerning the battery and the electric vehicle is too fluctuating. That’s why the use of KNN and other tested machine leaning models was a necessity. We clarified more this in the discussion section too. Please refer to the lines [636-645].

  • How does the proposed methodology handle uncertainties in renewable energy generation, such as variations in weather conditions?

This issue was tackled during the mathematical modeling of the used renewable energy generation. And the fact that our system used backup systems to avoid intermittency. This point was clarified in the new version of the paper. Please refer to the lines [656-663].

  • In real-world applications, what are the key factors influencing the practical implementation and adoption of the proposed hybrid renewable energy system in residential buildings?

Thank you very much for your vision and perspective on our approach. This remark has greatly contributed to our discussion section. Please refer to the lines [680-689].

Reviewer 2 Report

Comments and Suggestions for Authors

The work seems pretty good but I found some flaws:

The general objective of the paper is not clear, as is the novelty. Put it at the end of the introduction.

Is figure 8 necessary?

Have you thought about using the energy of the water in the pipes by placing microturbines?

In the literature there are alternatives to the use of this pressurized water.

Rodríguez-Pérez, Á. M., & Pulido-Calvo, I. (2019). Analysis and viability of microturbines in hydraulic networks: a case study. Journal of Water Supply: Research and Technology—AQUA, 68(6), 474-482.

The discussions section needs to be better.

The conclusions seem correct to me.

Author Response

Original Manuscript ID: sustainability-2845953

Original Article Title: Hybrid renewable production scheduling for a PV-Wind-EV-Batteryarchitecture using sequential quadratic programming and LSTM-KNN learning forsmart buildings

To: MDPI Sustainability

Re: Response to reviewers

Dear Editor, Dear Reviewer,

We appreciate the chance to revise and resubmit our work to address the concerns provided by the reviewers.

We have uploaded our detailed response to the remarks (the response to reviewers), along with a revised manuscript that includes highlighted revisions in yellow.

Best regards,

Asmae Chakir and Mohamed Tabaa

+------------------------------=

The work seems pretty good but I found some flaws:

The general objective of the paper is not clear, as is the novelty. Put it at the end of the introduction.

Thank you for you precious remark, we highlighted more our contributions. Please refer to the lines [123-126], [129-130] and [132-136].

Is figure 8 necessary?

Thank you for this question. Effectively, the figure 8 was eliminated. the text describing the approach is quite sufficient.

Have you thought about using the energy of the water in the pipes by placing microturbines?

It's a good prospect for the scalability of our approach, thank you for your proposal.

In the literature there are alternatives to the use of this pressurized water.

Rodríguez-Pérez, Á. M., & Pulido-Calvo, I. (2019). Analysis and viability of microturbines in hydraulic networks: a case study. Journal of Water Supply: Research and Technology—AQUA, 68(6), 474-482.

Thank you very much for your reference, it's very interesting. We also discussed its importance in the introduction. Please refer to the lines [51-54].

The discussions section needs to be better.

Thank you for your comments, several paragraphs have been added to make our discussion even better. Please refer to the lines [628-688].

The conclusions seem correct to me.

Thank you very much for your feedback.

Reviewer 3 Report

Comments and Suggestions for Authors

Thank you for providing an interesting article describing the solution to the problem of optimal management of the PV-Wind-EV-Battery architecture.

Although the article is extensive and complete, please consider making the following additions:

- supplementing Figure 1a with an element indicating the possibility of charging the battery via a PV system or Wind Turbine - as the texts (lines 455-457) show, such a possibility theoretically exists in the algorithm,

- in section 2.4 describing - Battery storage system, I do not find information on how the algorithm ensures not to over-discharge the battery,

- to bring the solution closer to the real implementation, it is worth taking into account the costs of energy from the electricity network, which are different at different times of the day - which may affect the energy management strategy.

Author Response

Original Manuscript ID: sustainability-2845953

Original Article Title: Hybrid renewable production scheduling for a PV-Wind-EV-Batteryarchitecture using sequential quadratic programming and LSTM-KNN learning forsmart buildings

To: MDPI Sustainability

Re: Response to reviewers

Dear Editor, Dear Reviewer,

We appreciate the chance to revise and resubmit our work to address the concerns provided by the reviewers.

We have uploaded our detailed response to the remarks (the response to reviewers), along with a revised manuscript that includes highlighted revisions in yellow.

Best regards,

Asmae Chakir and Mohamed Tabaa

+-----------------------------------------=

Thank you for providing an interesting article describing the solution to the problem of optimal management of the PV-Wind-EV-Battery architecture.

Although the article is extensive and complete, please consider making the following additions:

- supplementing Figure 1a with an element indicating the possibility of charging the battery via a PV system or Wind Turbine - as the texts (lines 455-457) show, such a possibility theoretically exists in the algorithm,

Please refer to the new version of the new version of figure 1-a. We added some elements that will help improving the understanding of the system.

- in section 2.4 describing - Battery storage system, I do not find information on how the algorithm ensures not to over-discharge the battery,

We appreciate your remark, we added equations in section 2.4 and 2.3 to show better how the algorithm ensures not to over discharge the battery. 

- to bring the solution closer to the real implementation, it is worth taking into account the costs of energy from the electricity network, which are different at different times of the day - which may affect the energy management strategy.

Thank you very much for your and perspective. This remark has greatly contributed to our discussion section. Please refer to the lines [680-689].

Round 2

Reviewer 2 Report

Comments and Suggestions for Authors

The authors have made all requested changes.